# Landslide Monitoring along the Dadu River in Sichuan Based on Sentinel-1 Multi-Temporal InSAR

**DOI:** 10.3390/s23073383

**Published:** 2023-03-23

**Authors:** Huibao Huang, Shujun Ju, Wei Duan, Dejun Jiang, Zhiliang Gao, Heng Liu

**Affiliations:** 1College of Water Resource and Hydropower, Sichuan University, Chengdu 610065, China; 2Guoneng Dadu River Hydropower Co., Ltd., Chengdu 610093, China; 3State Key Laboratory of Geohazard Prevention and Geoenvironment Protection, Chengdu University of Technology, Chengdu 610059, China; 4Institute of Software, Chinese Academy of Sciences, Beijing 100190, China; 5Faculty of Infrastructure Engineering, Dalian University of Technology, Dalian 116024, China

**Keywords:** JS-InSAR, SBAS, landslide, early identification, Dadu River, Luding

## Abstract

The Dadu River travels in the mountainous areas of southwestern China, one of regions with the most hazards that has long suffered from frequent geohazards. The early identification of landslides in this region is urgently needed, especially after the recent Luding earthquake (MS 6.8). While conventional ground-based monitoring techniques are limited by the complex terrain conditions in these alpine valley regions, space interferometric synthetic aperture radar (InSAR) provides an incomparable advantage in obtaining surface deformation with high precision and over a wide area, which is very useful for long-term and slow geohazard monitoring. In this study, more than 500 Sentinel-1 SAR images with four frames acquired during 2017~2022 were collected to detect the hidden landslide regions from the Jinchuan to Ebian Section along the Dadu River, based on joint-scatterer InSAR (JS-InSAR) and small baseline subset (SBAS) techniques. The results showed that our method could be successfully applied for landslide monitoring in complex mountainous regions. Furthermore, 143 potential landslide regions spreading over an 800 km area along the Dadu River were extracted by integrating the deformation measurements and optical images. Our study can provide a reference for large-scale geological hazard surveys in mountainous areas, and the InSAR technique will be encouraged for the local government in future long-term monitoring applications in the Dadu River Basin.

## 1. Introduction

The Dadu River is located in the transition zone between the Western Sichuan Plateau and the Sichuan Basin in China, near the edge of the Eastern Qinghai–Tibet Plateau [1]. The river flows through four cities in Sichuan in an L-shaped trend, with a total length of 1074 km. Due to the strong tectonic uplift movements, the topography of this area is mainly controlled by the geological structure. Where three major fault zones (Xianshuihe, Longmengshan and Anninghe) and other small faults are intersected and compounded, the Dadu River Basin exhibits the typical features of the intensely eroded alpine canyon landform, with complex natural geological conditions, rich natural resources and fragile ecological environments. Influenced by the intense regional neo-tectonic activity and large variation in precipitation, this region has become one of areas with the most hazards on southwestern China with a high incidence of various geohazards, including landslides, debris flows, earthquakes and so on. Among those, more than 1700 landslides spread all over the basin have been investigated by geological survey [2,3,4]. Moreover, the massive exploitation of mineral, forest and hydroelectric resources has promoted the occurrence of deep-seated landslides, threatening the safety of infrastructures and human life [5]. Therefore, the early identification of landslides in this region has become the focus of disaster prevention and mitigation for the local government, and constructing a map of geological hazards for the Dadu River Basin is of great significance to promote the efficiency of local geohazard monitoring and decrease the disaster losses.

The existing geohazard monitoring tools in the Dadu River Basin are mainly manual inspection, in situ measurement, photogrammetry and GNSS monitoring systems [6]. However, these methods are time-consuming and have very limited area coverage, and they are often impaired by heavy fog, cloud, rainfall and steep terrain. Deng et al. [7,8] investigated the landslide and valley evolution in the two ancient landslides of Luding District, and found that the geological structure and river erosion were the main contributors. Yan et al. [9] analyzed the reactivation and deformation characteristics of the Jiaju ancient landslide. Wang et al. [10] investigated the Moxi ancient landslide in Luding based on the simulation of landslide movements. Zhao et al. [11] assessed the deformation mechanism of the ancient landslide in Luding. Zou et al. [12] investigated the slope stability and landslide susceptibility all over the Dadu River Basin. Zhao et al. [13] provided insight on the large-scale landslide distribution and geological characteristics along the Sichuan–Tibet railway. The geological environment of the basin is complicated and densely vegetated, and the hidden landslide spots in this region are characterized by high concealment, wide distribution, inaccessibility and difficulties with investigation. As a result, the current field measurement methods cannot satisfy the actual needs of regular and continuous landslide monitoring with wide areas and high frequency in the Dadu River Basin. Landslide prediction has remained the key and difficult problem in geohazard prevention, and it has been strongly influenced by dynamics models, climate, river, complex geological environment and many other uncertain factors [14,15,16,17]. While large-scale surface deformation monitoring can provide new information for more reliable landslide prediction, new techniques are urgently needed to conduct landslide investigations and historical deformation process monitoring over the whole river basin.

With the development of space remote sensing techniques in the past few decades, interferometric synthetic aperture radar (InSAR) has shown unprecedented power in the identification and long-term observation of large-scale landslide disasters with an all-day and all-weather working capability [18]. Through the analysis of a series of interferometric phases, InSAR can obtain the surface deformation with high precision, high efficiency and a large spatial coverage at a low cost. Furthermore, the large number of satellite SAR systems with different bands and resolutions also provide abundant data sources for various geodetic applications. Among them, Sentinel-1 with Interferometric Wide Swath (IW), with its 250 km wide swath, small revisit time and global coverage, has greatly promoted large-scale deformation monitoring based on the InSAR technique [19]. Many researchers have applied the InSAR technique in landslide disaster detection in southwestern China and have achieved remarkable results. Li et al. [20] used InSAR to detect the stability of reservoir banks on the Jinshajiang River with a multi-sourced SAR dataset, and studied its relation with groundwater and rainfall. Zhang et al. [21] detected the landslide disasters in Wenchuan using stacking InSAR and small baseline subsets (SABS). Zhang et al. [22] studied the spatial distribution and controlling factors of wide-area landslides along the Sichuan–Tibet railway based on SABS. Chen et al. [23] used the GACOS-assisted stacking InSAR method to analyze landslides along Sichuan expressways with the combination of topographic and hydrological factors. Intrieri et al. [24] detected accelerating deformation behaviors prior to disasters in the Maoxian landslide based on the SqueeSAR method. However, the mountainous areas in southwestern China are mostly alpine and gorge regions. The complicated terrain, the dense coverage of vegetation and the high difference in elevation have brought great challenges to conventional InSAR processing methods, making the monitoring results unreliable due to the shadow, geometric distortion, decorrelation and coupling problems, especially for wide areas. As a result, few studies have investigated landslides over the whole Dadu River Basin. The lack of abundant landslide disaster information has largely influenced geohazard control and prevention work, consuming a lot of cost and time. 

Based on the considerations above, the combined joint-scatterer InSAR [25] and SBAS technique is presented in this work to study the potential landslide geohazards in the Dadu River Basin based on the collection of Sentinel-1 2017~2022 SAR data with multiple tracks. First, joint-scatterer InSAR preprocessing was performed on the multi-temporal coregistered SAR images with original resolution to improve the interferometric phase quality. Next, we used the SBAS InSAR method to generate the surface deformation map for wide areas. The landslide spots were subsequently detected, categorized and summarized according to the deformation time series results and regional geological features. Finally, the typical landslide regions were further analyzed and verified with GNSS measurements. Our results can help improve the existing monitoring system in the Dadu River, and the proposed method can provide new insights for geohazard prevention for the numerous river basins in southwestern China.

## 2. Materials and Methods

### 2.1. The Study Region

The Dadu River in Sichuan, China, mainly crosses Jinchuan, Danba, Luding and Shimian from north to south, and then eastward to Hanyuan, Ebian and Leshan. While most of the geohazards frequently occur from the Jinchuan to Ebian Section river basin, this work focused on the landslide investigation of this section, which is more than 800 km long. The strong dynamic process and vulnerable geological structure of this area has caused a large gap along the whole river. The region has a steep relief of terrain from high mountain valleys in the upstream to small hills in the downstream, with average altitude more than 5000 m and large height difference over 2000 m, as shown in Figure 1a. Additionally, from the Landsat false color image in Figure 1b, we can see that the land cover of the study area is densely vegetated along the Dadu River, which causes a severe decorrelation problem. Benefitting from many hydraulic resources, more than 20 cascade hydropower stations have been built or planned along the main stream. However, the river bank is deeply cut and eroded by flow with a sharp slope and broken rock mass, which can very easily trigger a geohazard. The frequent landslide disasters in the Dadu River Basin have caused many casualties and property damage, and they pose a great threat to local transportation and engineering construction, which has largely affected social and economic development. Furthermore, the Luding earthquake on 5 September 2022 further activated a large number of ancient landslides on the bank of the river, while many ancient landslide groups along the Dadu River have a high probability of sliding. As a result, the sole expressway lifeline of S211 is in danger, and some of the large-scale landslides may block the river, which will cause catastrophic implications for the cascade hydropower stations on the river. However, investigation of the hidden landslides over the whole Dadu River Basin has remained unclear, and recent studies mostly focus on small landslides in this region. Therefore, it has become extremely essential to conduct large-scale deformation monitoring with multi-temporal InSAR in the Dadu River Basin.

### 2.2. Data

The study region was fully covered by Sentinel-1A with C-band (5.6 cm) interferometric wide (IW) mode SAR images in four different frames. In total, 595 scenes of Sentinel-1A in ascending orbits were used in this processing with a time interval of 12 days, taken between October 2017 and October 2022, including 149 scenes for three frames of track 26 and 148 scenes for track 128. The parameter details are shown in Table 1, and the coverage of SAR images is shown in Figure 2. In addition, the external DEM was used to removed topographic phase and geocoding, and ALOS World 3D 30 m (AW3D30) DEM was selected in this processing. The precise orbit ephemerides (POD) files for Sentinel-1A were also downloaded and used to remove orbit error phase. Finally, GNSS measurements in typical regions were collected to confirm the effectiveness of InSAR results.

### 2.3. Method

The landcover of the study region is dominated by dense vegetation, thus conventional InSAR methods are greatly affected by spatial and temporal decorrelation problems in this region. In this study, we proposed the JS-InSAR and SBAS combined method to improve the deformation monitoring results in these mountainous areas of southwestern China. First, the single-look complex (SLC) images were generated from Setinel-1 SAR images with TOPS [26] mode SAR image coregistration method. Then, the SLC images were preprocessed using spatial adaptive filtering with JS-InSAR method, and the phase information of SAR images were reconstructed using an optimal estimator. In addition, SBAS InSAR was later used to obtain deformation rates through the analysis of a series of interferograms with small baselines. Finally, the landslide regions were extracted and analyzed based on deformation characteristics and visual interpretations. The flowchart of the whole processing chain is shown in Figure 3.

#### 2.3.1. JS-InSAR Preprocessing

JS-InSAR employs the distributed scatterer technique to focus on those temporal coherent ground targets, such as bare land and grassland, and mitigates interferogram decorrelation using joint-scatterer adaptive filtering for the statistically homogenous pixels (SHP). The SNR and phase quality for SAR images in the low-coherent regions can be greatly improved without influence on the permanent scatterers. Thus, this method can increase the spatial distribution density of measurements with high precision in non-urban areas. JS-InSAR mainly consists of three steps: joint-scatterer signal model, joint-scatterer goodness-of-fit test, joint-scatterer adaptive filtering.

(1)Joint-scatterer signal model

Unlike the SqueeSAR method [25], JS-InSAR performs joint processing on the spatial neighboring pixel stacks. As a result, JS-InSAR can identify more distributed scatterers regardless of pixel center shift or coregistration errors. For N time series of SLC images, given the joint data vector size with *k*_1_ × *k*_2_ (usually odd number, 3 × 3), the joint data vector **ux** can be written as follows:(1)$$ux\left(n\right)=[x{\left(n-\frac{k_{1}\times k_{2}-1}{2}\right)}^{T},\ldots ,x\left.(n{)}^{T},\ldots ,x{\left(n+\frac{k_{1}\times k_{2}-1}{2}\right)}^{T}\right],$$
$x(i),i\in (n-\frac{k_{1}\times k_{2}-1}{3},n+\frac{k_{1}\times k_{2}-1}{3})$ is the pixel stack, or a vector of complex values of the time series SLC images, i.e., $x\left(i\right)=\left\{x_{1}\left(i\right),x_{2}\left(i\right),\ldots \ldots ,x_{N}\left(i\right)\right\}$, $x_{j}\left(i\right)$ is the *i*th pixel in *j*th SLC. While the middle pixels **x**(*n*) are the processing units in this calculation, the phase values for **x**(*n*) are updated in the JS-InSAR processing. The joint data vector is shown in Figure 4.

(2)Joint-scatterer goodness-of-fit test

The JS goodness-of-fit test was used to identify SHP around the center pixel and evaluate the similarity of joint scatterers. As the traditional goodness-of-fit methods cannot satisfy the needs of multi-dimensional JS vectors, a two-step two-sample KS test was used to calculate the statistical correlation of JS vector data. The two-sample KS test was applied on the spatial and temporal domain, respectively. First, in the spatial dimension, the maximum patch similarity of the center pixel was defined as the KS test result of two JS vectors for every SLC image. Then, in the time dimension, the KS test calculation on the pixel stack was the same as the conventional method, while the single pixel value was replaced by the spatial maximum patch similarity of each SLC. The KS test was given as follows:(2)$$D_{KS}=sup|P_{ij}(x)-P_{center}(x)|,$$

The $P\left(x\right)$ is the cumulative amplitude distribution of scatterers in a patch. The KS test result refers to the maximum difference in cumulative distribution functions of two vectors, which can evaluate the similarity between two joint scatterers. The operation was performed on all of the pixels in the window of every center pixel, and JS-InSAR identified more distributed scatterers. 

(3)Joint-scatterer adaptive filtering

The ground objects for the neighboring distributed scatterers have the similar scatter mechanism, thus they share the same spatial–temporal decorrelation effect. Through the statistical characteristics analysis of these SHP, the real phase value of the center pixel could be estimated using joint-scatterer adaptive filtering. For better use of the phase values of SHP in the window of the center pixel, the similarity between SHP and center pixel calculated using the KS test was used as the weight function $w\left(i\right)$, which is given as follows:(3)$$w\left(i\right)=e^{-\frac{D_{KS}(i)}{\alpha }},$$
while the smaller value of $D_{KS}\left(i\right)$ means the more similar of two joint scatterers, it will contribute more to the covariance. The joint covariance matrix can be written as follows:(4)$$C_{ux\left(n\right)}=\frac{1}{\sum_{l=1}^{\Delta }w\left(l\right)}\sum_{l\in \Delta }w\left(l\right)ux\left(l\right)ux^{H}\left(l\right),$$
where △ is the set of joint vector data, or SHP candidates. Then, the phase optimization can be determined using phase triangulation [26] or eigen decomposition for the joint covariance matrix, and the reconstructed phase will replace the original value of SLC images. Thus, the coherence of SAR images can be greatly improved, and the decorrelation noise of the interferometric phase will be reduced considerably. In the practical processing, the SHP search and spatial filtering window size was 7 × 7 with the consideration of SHP numbers and computation burden.

#### 2.3.2. SBAS Processing

The SBAS technique has been widely used in landslide monitoring, and can acquire deformation time series for complex non-urban regions at the sacrifice of spatial and time resolution [27]. After JS-InSAR preprocessing of SLC images, SBAS time series InSAR processing can further reduce the effect of decorrelation and topographic influence by multilooking and limiting interferograms with small temporal and spatial baselines. It starts with forming interferometric pair networks based on the given perpendicular and temporal baseline threshold, and the interferogram coherence can also be taken into consideration. Then, the multilook interferograms are generated and flattened with topography phase removed by external DEM. The generated differential interferograms are unwrapped using the MCF phase unwrapping method [28]. To obtain more accurate deformation results, those unwrapped interferograms with low phase quality are removed and the interferometric pairs are updated. The unwrapped phase for all of the interferograms can be given as follows:(5)Aφ=δφ,
for N SLC images and M interferograms, the design matrix *A* will be M × N with only two elements 1 and −1 for each row, which define the interferometric pair relation. *δφ* is the unwrapped phase vector for all of the interferograms with size of M × 1. Before retrieving the real phase time series *φ* by inversion, the deformation rate and residual topography are estimated based on the linear relation between the phase value and the baseline. Then, the linear deformation phase and topography phase are removed from unwrapped phase, and the phase time series with single reference are obtained by residual unwrapped phase inversion with SVD inversion method. Finally, the residual phase time series are atmospherically filtered and converted into LOS displacement time series. Only the coherent pixels with high temporal coherence of residual phase can be seen as the reliable measurements. In the practical Setinel-1 SAR image processing, the multilook factor of 10 (range) and 2 (azimuth) were used in the interferogram generation, while the time and spatial baseline was set to 72 days and 200 m, respectively. An example baseline map of SAR dataset with track 26-2 is shown in Figure 5. The spatial filtering window is 16 × 16, and the temporal coherence threshold is 0.7.

## 3. Results and Analysis

### 3.1. JS-InSAR Optimization Results

To validate the effectiveness of the JS-InSAR method, we compared the interferogram and coherence between the original and the optimized phase. The interferograms of the two interferometric pairs for the dates 30 October 2021 vs. 23 November 2021 and 22 May 2022 vs. 3 June 2022 with track 26-2 were selected, and the parts of the interferometric phase are shown in Figure 6. We observed that the phase noise of the interferometric phase (without adaptive filtering processing) was largely reduced with JS-InSAR filtering, and the quality of the interferograms were greatly improved, especially the regions near the Dadu River. The optimized interferograms could lower the difficulty and increase the reliability of phase unwrapping in the SBAS processing. To further illustrate the improved quality of the interferograms, the average coherence for every interferogram was calculated and counted, and can be seen in Figure 7. The optimized interferograms showed higher coherence than the original interferograms in general, and thus the JS-InSAR phase optimization method was proven to be effective.

To further verify the performance of our method, we compared the deformation rate distribution of the proposed method with the SBAS method as shown in Figure 8. We observed that the deformation range for the conventional SBAS was larger than our method, which meant that the SBAS method had obtained more extreme deformation values. Furthermore, there were evident deformation errors (the red box in Figure 8b) caused by incorrect phase unwrapping in the SBAS method. Thus, the proposed method could acquire more reliable deformation results than the conventional results.

### 3.2. The Deformation Rate Map of InSAR Results

The annual average deformation rate for the Dadu River Basin from October 2017 to October 2022 was obtained using the SBAS method, as shown in Figure 9. The InSAR measurements in the figure were the high coherent scatterers in the SBAS processing, and the deformation results were along the line of sight (LOS) direction with color coding based on its values, while the positive or negative value represented surface displacement moving towards or away from the satellite, respectively. From the figure, we observed that the JS-InSAR method had extracted more ground targets in a complex environment, with the average point density of 1496/km^2^, thereby obtaining clearer and more accurate displacement information. Moreover, no obvious large-scale atmospheric or orbit errors were found in the deformation results, avoiding deformation overestimation and artifacts. The average deformation rate of the Dadu River Basin for 2017~2022 was −87.5~36.7 mm/year, with 85% deformation in regions located in the range between −10~10 mm/year, which proved that most of the areas in the Dadu River maintained a certain degree of stability. We also found that some landslide groups with large deformation mainly occurred in Danba, Luding and Shimian. However, some slopes along the river bank were affected by the SAR imaging geometry and alpine valley topography and could not be fully investigated due to the shadow and decorrelation problems.

The displacement map of the three major deforming regions was enlarged and shown in Figure 10 for the (a) Danba Section, (b) Luding Section and (c) Shimian Section, and a range of suspected landslides with noticeable subsidence could be clearly seen in these areas. In the Danba Section, a large number of landslides with an average deformation rate of more than -30 mm/y could be identified along the Dadu River, such as the Jiaju landslide [29], Danba landslide, Bawang landslide and so on. The most serious deformation area in this region was situated in the ancient landslide of Gezhong village, and the cumulative displacement from 2017 to 2022 was more than −220 mm. The Luding district was strongly influenced by intense geological activity, where the most geohazards could be found. The maximum subsidence of the Luding Section was located in the Weishe village, which had reached about −147 mm from 2017 to 2022. The Shimian Section is in the lower part of the Dadu River and covered by dense vegetation. Thus, most of the detected landslides were not close to the Dadu River. The maximum deformation area was in the Baojiawuji village, and its subsidence was −131 mm during 2017~2022.

### 3.3. Landslide Extraction and Analysis

To investigate the geohazard distribution along the Dadu River and its controlling factor in detail, major landslides in the 5 km buffer of the Dadu River were extracted based on the average deformation rate of the Dadu River Basin from October 2017 to October 2022, with the combination of Google optical images. In our experience, the obvious deforming regions with the following conditions were considered as hidden landslide spots: the maximum subsidence rate was more than −30 mm/y; the maximum cumulative displacement was more than 80 mm; the slope angle was more than 15°; the total deforming areas were more than 0.01 km^2^. Furthermore, high coherence, topographic trend and vegetation cover were also taken into consideration to avoid potential decorrelation and unwrapping errors. A total of 143 evident landslides were detected, which greatly threaten the Dadu River, as shown in Figure 11.

From Figure 10, we observed that the spatial distribution of landslides in the Dadu River Basin was uneven at different parts of the river. In the upstream of the river, this region belonged to the Ganzi Aba crease southeastern zone with a cold and dry plateau climate, and the emergence stratums were mainly intrusive granite. Thus, most of the areas in this region seemed to be very stable, such as the Jinchuan Section. However, the topography of the alpine valleys and river erosion cutting had high potential to cause slope instability in the Danba Section, where a large number of ancient landslide groups were distributed. Furthermore, in the Luding Section of the middle stream, earthquakes, faults and other geological activities were very typical and active in this region. These have become important contributing factors of landslide geohazards, as can be seen from the yellow lines in the figure. In the downstream of the Dadu River, the plentiful rainfall and extreme weather in the Shimian and Hanyuan Sections most likely resulted in the frequent occurrence of large-scale landslides. Thus, the landslide distribution in the Dadu River exhibited close relations with the geological structure and climate features.

According to the detected landslide inventory shown in Figure 10, we studied two typical landslides in Danba and Luding, respectively, to illustrate the landslide body based on their cumulative time series displacement, and both of which had evident deformation trends, as shown in Figure 12. It can be seen in Figure 12a that all three points had the continuous deformation trend in the slope of Danba. The maximum subsidence, P3, was located in the middle of the slope, which was about −120 mm from 2017 to 2022, while the top of the slope had relatively smaller deformation in points P1 and P2. From the optical images, we learned that the road construction and river cutting had been the major causes of slope instability. The same situation applied to the second landslide in Luding, which had a larger deformation rate due to the steeper slope with a subsidence of −120 mm. The P1 point at the top of the slope remained stable compared to P2 and P3. As the two landslides were closely adjacent to the Dadu River, if a landslide occurred, they might block the river and endanger the cascade hydropower station in the downstream, similar to the lake barrier after the Luding earthquake. The local government should pay attention to these detected regions. Therefore, despite the low resolution of Sentinel-1 with the SBAS method, our proposed method could be applicable in landslide analysis through long-term series monitoring.

### 3.4. Validation of InSAR Results

Field investigations in the Dadu River were conducted very early for some specific areas, and we collected three GNSS points, ZP01, ZP02 and ZP03, from the Zhengjiaping landslide in this validation. The three-dimensional measurements of GNSS with the same duration of SAR image acquisition were first projected into the InSAR line-of-sight direction, and the difference in the initial deformation offsets between two methods were removed. The InSAR time series displacement and GNSS measurements from 1 October 2017 to 30 October 2022 were then compared at the three points, as shown in Figure 13.

From Figure 13, it can be clearly seen that the InSAR displacement time series were highly consistent with the GNSS measurements. The three measurements shared very similar deformation trends with the InSAR results from 2017 to 2022. While ZP01 had the highest agreement with the two methods, the InSAR time series exhibited more fluctuation in ZP02 and ZP03, which might be caused by the scatterer near the vegetation cover or different geo-locations between the two points. The RMSE of the difference between the GNSS and InSAR time series for the three stations were 0.5 cm, 0.71 cm and 0.86 cm, respectively. The results showed that our proposed time series InSAR method was reliable, but the accuracy of the InSAR results were still impacted by the processing errors.

## 4. Discussion

The landslide detection and deformation results in the previous section were consistent with previous studies, especially in the typical landslide regions, such as Danba, Jiaju village, Gezhong village, Zhenjiaping and Luding, while many newly detected landslides should be further inspected by field investigations. Due to the SAR side-looking image geometry, the mountainous areas with steep topographic relief in the Dadu River Basin caused a serious geometric distortion problem. A large number of slopes along the river had been missed in the monitoring because they were totally covered by shadows and did not have adequate measurement points in the InSAR processing. Moreover, the InSAR deformation results in some areas with extremely dense vegetation were still affected by strong decorrelation errors, and these unreliable measurements might bring about deformation artefact in the landslide interpretation. Finally, the number of the Sentinel-1 SAR images with acquisition time after the Luding earthquake were not enough to investigate the influence of the earthquake striking the study region. Thus, future research will incorporate more Sentinel-1 images into the time series analysis. The Sentinel-1 datasets with both ascending and descending orbits will be combined to conduct a more comprehensive landslide survey for the Dadu River Basin.

## 5. Conclusions

In this study, the JS-InSAR and SBAS combined method was proposed to acquire surface deformation in the mountainous areas of southwestern China. The JS-InSAR preprocessing was introduced to improve the quality and coherence of the interferograms through the joint-scatterer optimal phase estimation, thus obtaining better deformation results compared to the conventional SBAS processing. A total of 595 sentinel-1 SAR images taken between October 2017 and October 2022 were used to conduct landslide monitoring in the Dadu River from the Jinchuan to the Ebian Section. The annual average deformation rate map in the LOS direction was generated based on the proposed method, and the deformation rate of the Dadu River Basin was between −87.5 and 36.7 mm/year from 2017 to 2022. It was found that the large-scale landslide groups were mainly concentrated in Danba, Luding and Shimian County. With the comprehensive consideration of the deformation rate map, geological structures and optical images, a total of 143 evident landslides along the Dadu River were detected by visual interpretation. The landslide inventory of our results can provide useful information for local government about geohazard mitigation. The spatial distribution of the detected landslides was strongly correlated with their geological and climate characteristics, and showed significant differences between upstream and downstream. In addition, the deformation time series of two typical landslides in Danba and Luding were further taken into analysis, and the reliability of the landslide monitoring results could be well validated. Finally, the comparative results of InSAR and GNSS in the Zhengjiaping landslide demonstrated the accuracy and effectiveness of the proposed method. Thus, the application of the multi-temporal InSAR and Sentinel-1 dataset can greatly improve the level of safety monitoring in the Dadu River. The InSAR results in this work will help local government to build a geohazard map for the whole Dadu River Basin, and the detected landslide groups can be directly applied in geohazard management. Furthermore, our proposed method has been proven very useful for landslide monitoring in the complex geological environment of southwestern China, and it can improve the efficiency of geohazard prevention by revealing geohazard evolution and discovering potential hazards, which will facilitate the “Early detection, early treatment” goal of geohazard mitigation and control.

## Figures and Tables

**Figure 1 sensors-23-03383-f001:**
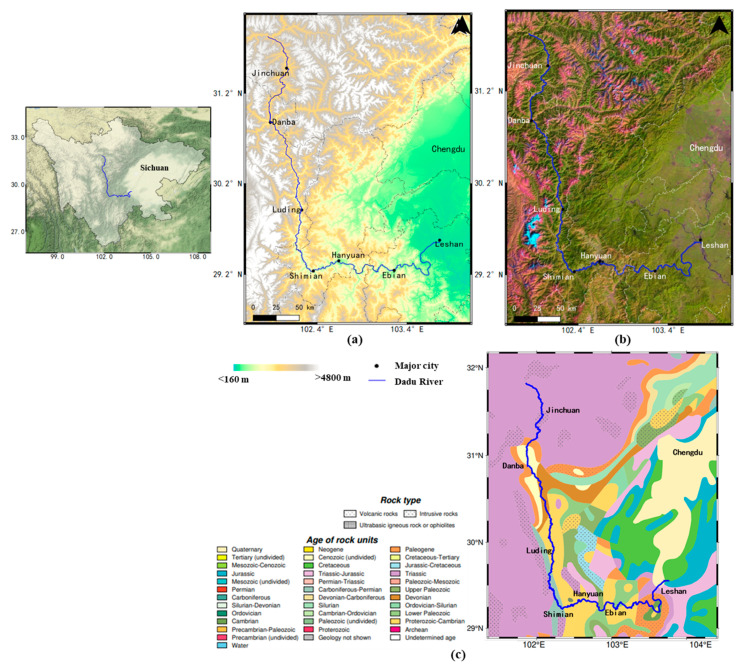
The study region. The blue line is the Dadu River. (**a**) is the topographic image. (**b**) is the Landsat false color image for 2021. (**c**) is the geological map of the Dadu River. The right figure is the location of the Dadu River in Sichuan.

**Figure 2 sensors-23-03383-f002:**
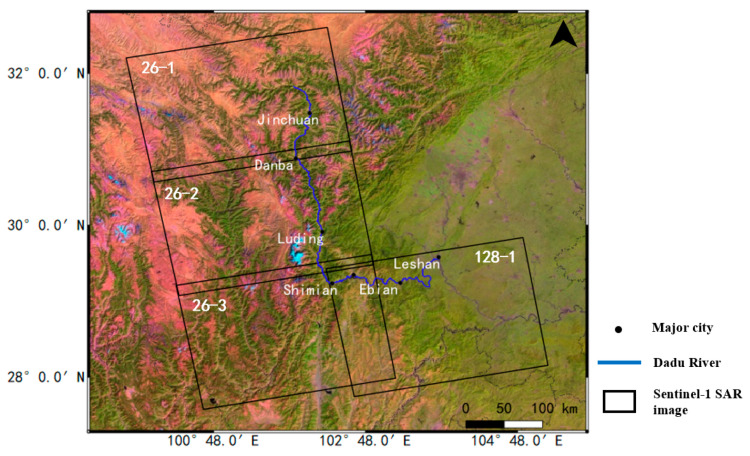
The location of Sentinel-1A dataset coverage (black box).

**Figure 3 sensors-23-03383-f003:**
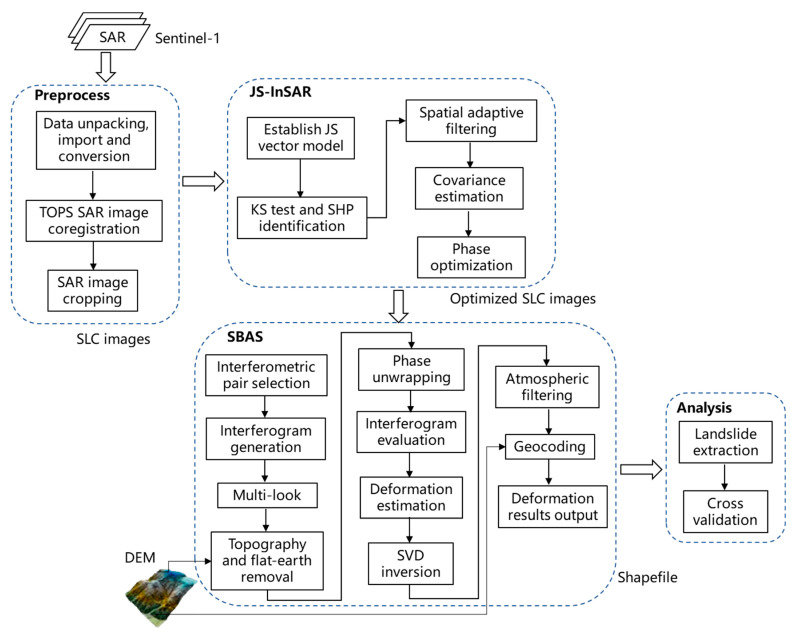
The processing chain of the proposed method.

**Figure 4 sensors-23-03383-f004:**
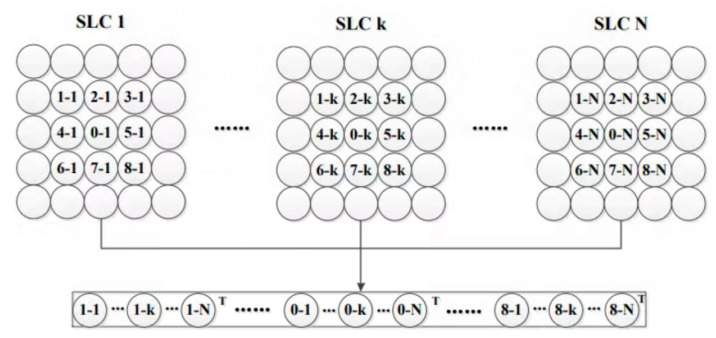
Joint-scatterer model for window with 3 × 3.

**Figure 5 sensors-23-03383-f005:**
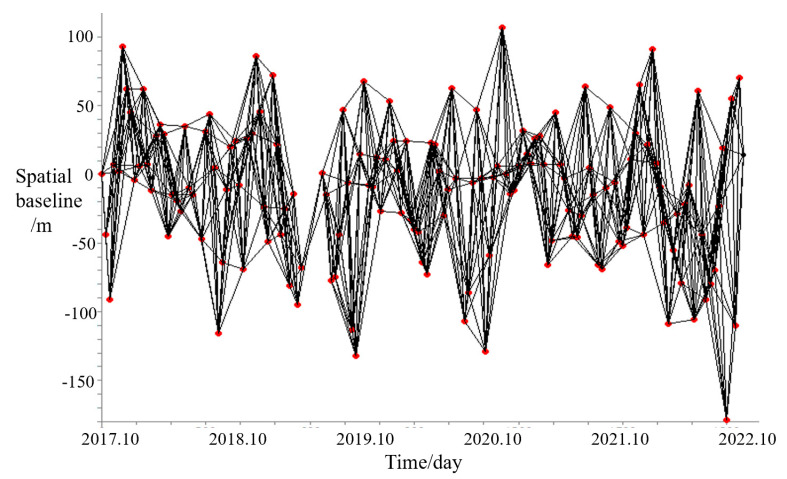
Spatiotemporal baseline of the Sentinel images with track 26−2.

**Figure 6 sensors-23-03383-f006:**
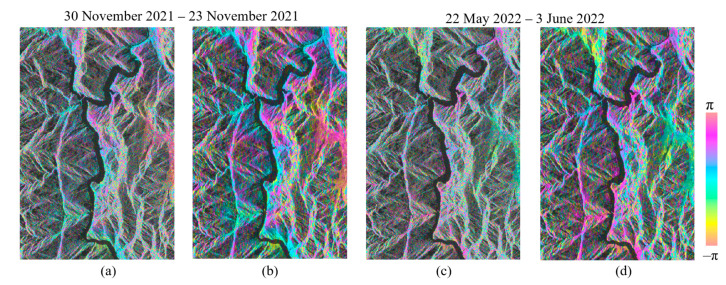
The differential interferograms with 10:2 multilook for parts of the track 26-2 with the interferometric pair of 30 October 2021 vs. 23 November 2021 and 22 May 2022 vs. 3 June 2022 in SAR coordinate. (**a**,**c**) Original phase, (**b**,**d**) JS-InSAR optimized phase.

**Figure 7 sensors-23-03383-f007:**
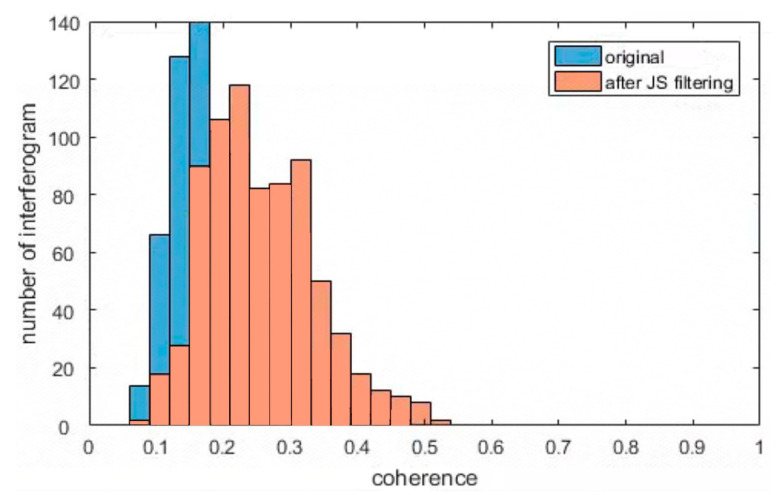
The average coherence comparison of the interferograms before and after JS-InSAR processing.

**Figure 8 sensors-23-03383-f008:**
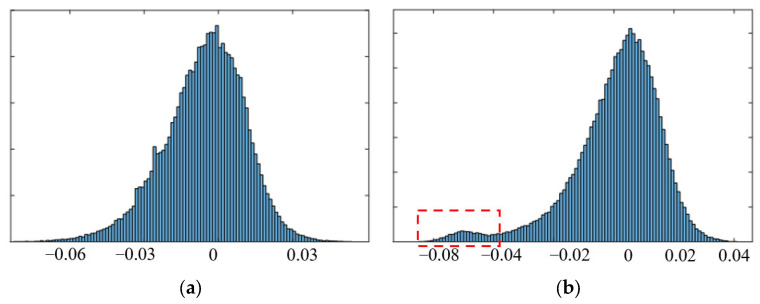
The histogram of the deformation results for (**a**) the proposed method and (**b**) SBAS.

**Figure 9 sensors-23-03383-f009:**
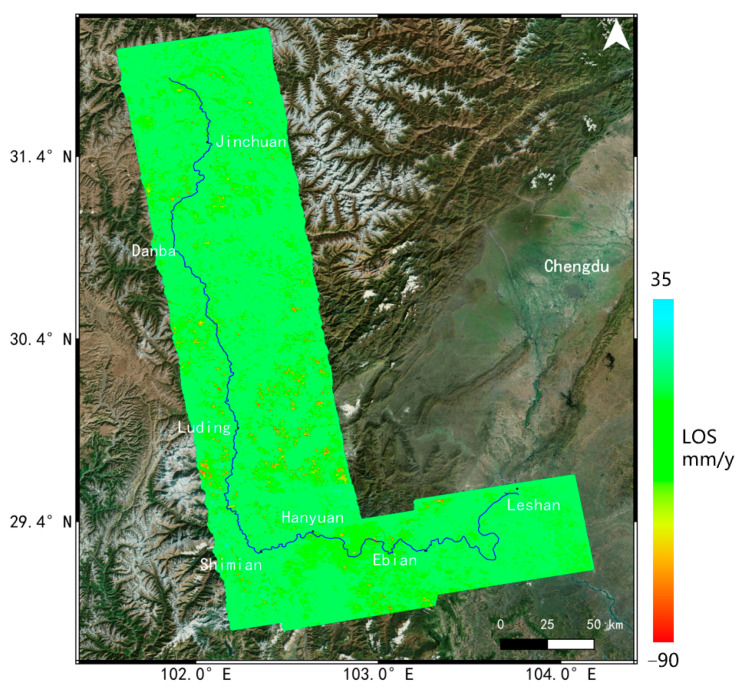
The deformation rate map for the Dadu River Basin from −2017 to 2022. The blue line is the Dadu River.

**Figure 10 sensors-23-03383-f010:**
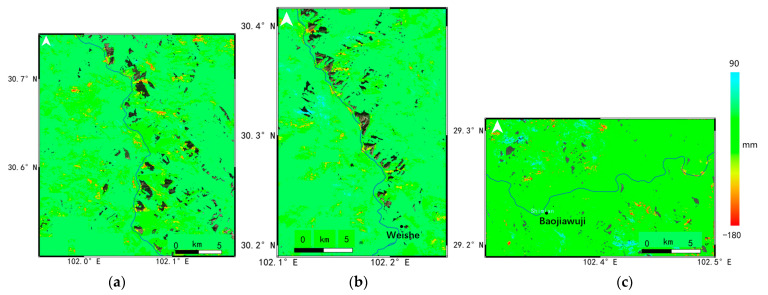
The displacement map for the Danba (**a**), Luding (**b**) and Shimian (**c**) Section (The blue line is the Dadu River).

**Figure 11 sensors-23-03383-f011:**
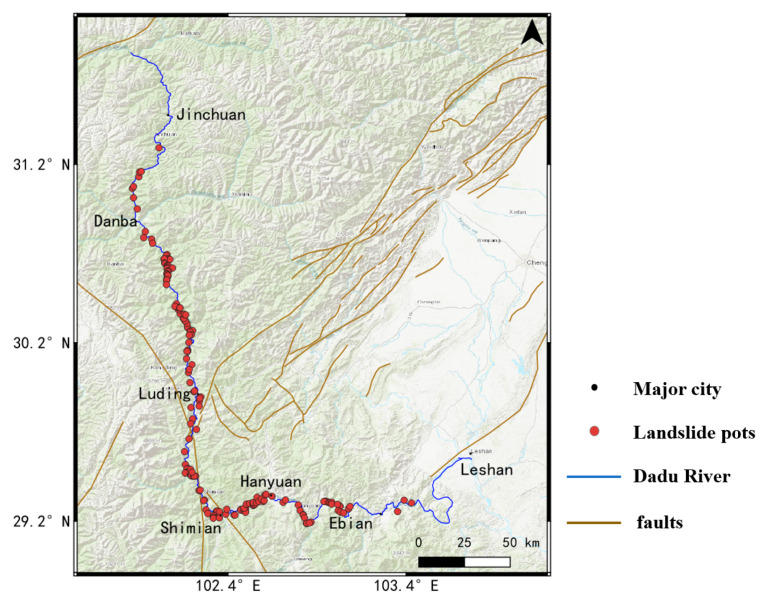
The distribution of landslides (red dot) along the Dadu River detected using InSAR results. The yellow lines are the geological faults in the Dadu River Basin.

**Figure 12 sensors-23-03383-f012:**
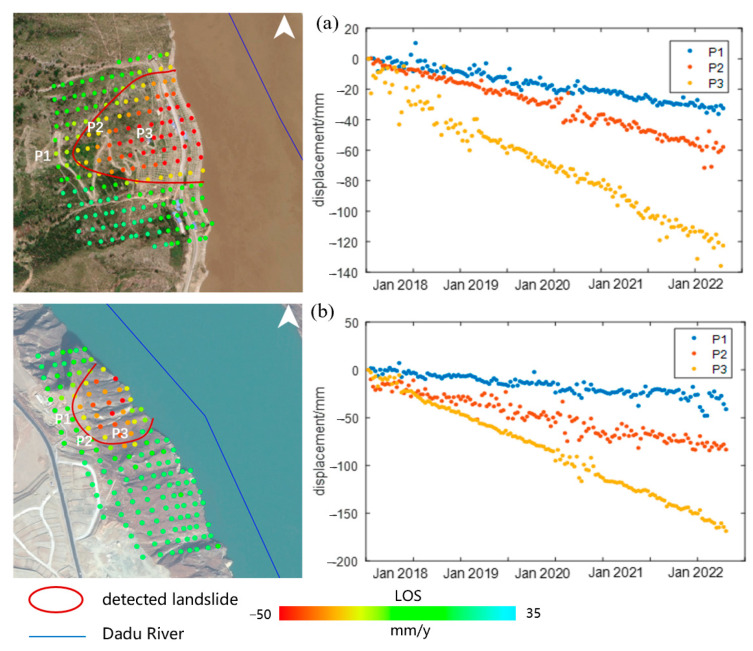
The time series displacement for two typical landslides detected in Danba (**a**) and Luding (**b**), respectively.

**Figure 13 sensors-23-03383-f013:**
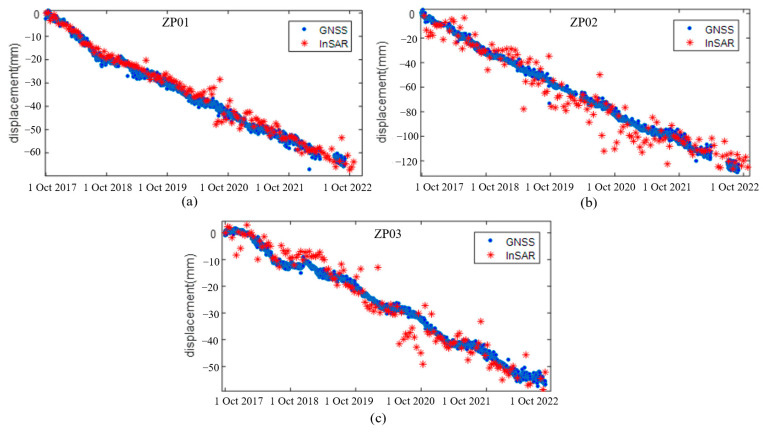
The comparisons between displacement time series and GNSS measurements of the Zhengjiaping landslide in the LOS direction for three stations: ZP01 (**a**), ZP02 (**b**) and ZP03 (**c**), respectively.

**Table 1 sensors-23-03383-t001:** Sentinel-1 SAR data used in the study.

Orbit-ID	Direction	Polarization/Mode	Angle	Resolution(m) Azimuth × Range	Date	Image Number
26-1	Ascending	VV/IW	43.9°	13.9 × 2.3	3 October 2017~25 October 2022	149
26-2	Ascending	VV/IW	43.9°	13.9 × 2.3	3 October 2017~25 October 2022	149
26-3	Ascending	VV/IW	43.9°	13.9 × 2.3	3 October 2017~25 October 2022	149
128-1	Ascending	VV/IW	33.9°	13.9 × 2.3	10 October 2017~20 October 2022	148

## Data Availability

The Sentinel-1 datasets can be acquired freely from Copernicus and ESA, https://search.asf.alaska.edu/#/ (accessed on 21 March 2022).

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
