# Peer review of "Landslide Monitoring along the Dadu River in Sichuan Based on Sentinel-1 Multi-Temporal InSAR"

_sensors, 2023, doi:10.3390/s23073383_

Round 1

Reviewer 1 Report

The authors used the joint method of JS-InSAR and SBAS-InSAR to generate the deformation along the Dadu river with 595 sentinel-1A datasets. The results of this paper can provide useful information for the prevention of geologic hazards. However, the author executed the multilook operation after the JSInSAR, leading to the loss of the advantage of JSInSAR which can obtain deformation details with single look resolution. Additionally, direct multilook operation of traditional SBAS-InSAR with appropriate interferograms can also lead to improvement of coherence and phase quality, how can you separate the contributions between the two methods? Moreover, the results are analyzed based on the potential landslides, however, the detection method for these potential landslides is not provided. Furthermore, I think there is still some room for improving the language of this manuscript.

Some other problems must be explained as follows,

1.      In the ‘Introduction’ part, many existing studies related to the deformation monitoring of the Dadu river basin can be found, however, the authors do not provide a review of these papers, leading to the contribution of this paper is not unclear.

2.      In the ‘Materials and Methods’ part, P8, L223, which method do you use to refine the interferograms? Have you considered the seasonal variation of vegetation in mountainous areas? Moreover, the authors described the geological environment of the study area, please add the geological strata map.

3.      P10, L277, you said the results in this paper can obtain more clearer and accurate displacement, which method do you use as a reference?

4.      In section 3.3, the authors analyzed the causes of these potential landslides, however, these reasons are only possibilities and not well supported by evidence, please reorganized these sentences.

5.      In section 3.4, the authors said ‘the difference of the initial deformation offsets between two methods are removed’, does it mean you have translated the relative values of InSAR to the absolute values of GNSS, if yes, which GNSS station have you chosen? As shown in Figure 10, the author did not select any of the three sites as reference GNSS stations. Moreover, please give the quantitative analysis between InSAR and GNSS.

Additionally, some minor problems should be corrected.

1.      P2, L58, the punctuation should be corrected.

2.      Figure 2 can be combined with Figure 1(b).

3.      In section 3.2, please explain the definition of the positive and negative values in Figure 8. The signs of the velocity need to be unified throughout the text.  

4.      P10, L289, the 30 mm/yr is positive values or the absolute values of the deformation velocity?

5.      P10, L295-298, the location ‘Weishe village’ and ‘Baojiawuji village’ cannot be found in Figure 9.

6.      P12, L322, I cannot find the ‘yellow lines’ in Figure 10.

Author Response

Thank you for your comment, and please see the attachment.

Reviewer 2 Report

Please find my report below.

Author Response

  1. Please add JS-InSAR and SBAS method to the keywords;

Response: Thank you very much for your comment. We have added and changed the keywords as follows:

“ Keywords: JS-InSAR; SBAS; Landslide; Early Identification; Dadu River; Luding”

  1. In page 5, line 149, please define TOPS and add reference.

Response: Thank you very much for your comment. We have added the reference as follows.

“Yagüe-Martínez N, Prats-Iraola P, Gonzalez F R, et al. Interferometric processing of Sentinel-1 TOPS data. IEEE transactions on geoscience and remote sensing, 2016, 54(4): 2220-2234.”

  1. In page 6, line 168, please define the SqueeSAR method and add a reference..

Response: Thank you very much for your comment. We have added the reference as follows.

“Unlike SqueeSAR method [25], JS-InSAR performs jointly processing on the spatial neighboring pixel stacks.”

“Ferretti A., Fumagalli A., Novali F., et al. A new algorithm for processing interferometric data-stacks: SqueeSAR. IEEE transactions on geoscience and remote sensing, 2011, 49(9): 3460-3470. ”

Reviewer 3 Report

This study aims to detect landslides along the Dadu River in Sichuan utilizing Sentinel-1 SAR images. While the topic is interesting, the manuscript has several issues. I would like the authors to address my following comments diligently before the manuscript may be accepted for publication.

·         The background of this study has been developed primarily based on the context of the study area. For instance, in lines 50-60, the authors explained the existing geohazard monitoring tools in the Dadu River basin. Also, they evaluated the drawbacks of those tools. However, it is equally important to investigate geohazard monitoring systems in different regions of the world. Therefore, I would suggest the authors revise the introduction section to highlight the major gaps in the existing literature in order to clarify the contribution of this study.  

·         The existing studies on landslides are predominantly focused on predicting landslide events based on some ground observations. The introduction section could be further improved by explaining this aspect of landslide studies. Besides, predicting landslides could be subject to uncertainties. The authors could explain various sources of uncertainties regarding landslide predictions. They could consult the following articles to improve the motivation of this study.

https://doi.org/10.3390/rs12203347

https://doi.org/10.1016/j.geomorph.2009.06.020

https://doi.org/10.1016/j.geomorph.2010.09.004

https://doi.org/10.1016/j.geomorph.2006.04.007

·         Section 2.1: The authors need to justify the choice of this study area.

·         Figure 1 needs to be developed more professionally. The study area boundary is not clear in the bottom maps.

·         2.3. Method: this study applied a combined JS-InSAR and SBAS methods. Was a similar method used elsewhere in the world? If so, the authors need to cite those.

·         Section 3.4 explains the validation techniques. However, some of the texts should be part of the methodology. Section 3.4 should mainly focus on explaining the validation results.

·         The manuscript does not have a discussion section. I would suggest the author include a discussion section in order to critically evaluate their results in light of the existing studies.

·         What are the potential sources of uncertainties in the results? How were the uncertainties treated in this study?

·         I feel the results of this study could be further utilized in landslide susceptibility and hazard studies. The authors could explain the theoretical and policy implications of this study.

Author Response

(The authors gave the same response as above.)

Round 2

Reviewer 1 Report

The authors modified the manuscript accordingly, however, some problems are not explained well and still need to improve.

1.      The authors said that landslide detection is based on visual interpretation with the assistance of optical images and InSAR results, however, they do not provide detailed information, for example, the velocity and slope threshold adopted in this paper. The detection results can directly affect the following discussion result.

2.      The authors said that the combined method can obtain better interferograms compared to the conventional SBAS and PSInSAR methods, however, they do not provide the point density of these two methods, and just directly give the conclusion.

3.      The authors explained the comparison between GNSS and InSAR results, however, I still think the comparison is not correct. First, they do not provide the reference GNSS station which can lead to the failure of comparison between relative value (InSAR results) and absolute value (GNSS). Usually, the RMSE of the reference GNSS station is approximately 0, however, I cannot find the reference GNSS station in the revised manuscript. Second, they do not provide the RMSE of each GNSS station, the description added in the revised version ‘The RMSE of the displacement difference between GNSS and InSAR time series is less than 0.86cm’ is not clear.

Author Response

Thank you for your comment. Please see the attachment.

Reviewer 3 Report

The authors have adequately addressed my comments. I recommend publication of the revised manuscript. 

Author Response

Thank you for your comment!